# Repurposing Sulfasalazine as a Radiosensitizer in Hypoxic Human Colorectal Cancer

**DOI:** 10.3390/cancers15082363

**Published:** 2023-04-18

**Authors:** Lisa Kerkhove, Febe Geirnaert, Amir Laraki Rifi, Ka Lun Law, Adrián Gutiérrez, Inge Oudaert, Cyril Corbet, Thierry Gevaert, Inès Dufait, Mark De Ridder

**Affiliations:** 1Department of Radiotherapy, Universitair Ziekenhuis Brussel, Vrije Universiteit Brussel, 1090 Brussels, Belgium; lisa.kerkhove@vub.be (L.K.); ines.dufait@vub.be (I.D.); 2Department of Hematology and Immunology, Myeloma Center Brussels, Vrije Universiteit Brussel, 1090 Brussels, Belgium; 3Pole of Pharmacology and Therapeutics (FATH), Institut de Recherche Expérimentale et Clinique (IREC), UCLouvain, 1200 Brussels, Belgium

**Keywords:** hypoxia, sulfasalazine, redox homeostasis, ferroptosis, radiotherapy

## Abstract

**Simple Summary:**

Radiotherapy is a cornerstone for the treatment of colorectal cancer. Tumor cells present in an environment lacking oxygen (hypoxia) are resistant, leading to patient relapse. Altering redox homeostasis and inducing cell death of hypoxic cancer cells is a promising strategy to overcome radioresistance. In this study, redox homeostasis was targeted, and cell death (ferroptosis) was induced in colorectal cancer cells by treating them with the FDA-approved drug sulfasalazine. Overall, sulfasalazine treatment improved the response to radiotherapy in a model system of human colorectal cancer cells.

**Abstract:**

xCT overexpression in cancer cells has been linked to tumor growth, metastasis and treatment resistance. Sulfasalazine (SSZ), an FDA-approved drug for the treatment of rheumatoid sarthritis, and inflammatory bowel diseases, has anticancer properties via inhibition of xCT, leading to the disruption of redox homeostasis. Since reactive oxygen species (ROS) are pivotal for the efficacy of radiotherapy (RT), elevated levels of ROS are associated with improved RT outcomes. In this study, the influence of SSZ treatment on the radiosensitivity of human colorectal cancer (CRC) cells was investigated. Our principal finding in human HCT116 and DLD-1 cells was that SSZ enhances the radiosensitivity of hypoxic CRC cells but does not alter the intrinsic radiosensitivity. The radiosensitizing effect was attributed to the depletion of glutathione and thioredoxin reductase levels. In turn, the reduction leads to excessive levels of ROS, increased DNA damage, and ferroptosis induction. Confirmation of these findings was performed in 3D models and in DLD-1 xenografts. Taken together, this study is a stepping stone for applying SSZ as a radiosensitizer in the clinic and confirms that xCT in cancer cells is a valid radiobiological target.

## 1. Introduction

Sulfasalazine (SSZ), an FDA-approved drug for the treatment of rheumatoid arthritis and inflammatory bowel diseases (IBD), is a potent xCT inhibitor and has attracted a lot of interest to be repurposed as an anticancer drug. SSZ already showed anticancer effects in a range of different tumor types, including colorectal cancer (CRC) [1].

There are two main mechanisms that are widely accepted to drive the observed anticancer effects of SSZ. The most reported mechanism is a disturbance of redox homeostasis. High levels of reactive oxygen species (ROS) are responsible for oxidative damage leading to cell death. Thus, it is to the cancer cell’s advantage to retain the intracellular ROS levels under a non-toxic threshold. This results in a tight balance between ROS and antioxidant (AO) enzymes in order to maintain a state of redox homeostasis [2,3,4]. SSZ disturbs this balance by inhibiting the light subunit of system xC−, xCT. System xC− is an antiporter responsible for the intracellular uptake of cystine coupled to the extracellular release of glutamate [5,6]. Cysteine is the rate-limiting factor of glutathione (GSH), one of the most abundant AO of a cell, and is produced by the reduction of intracellular cystine, of which the availability is dependent on system xC− [5]. Consequently, treatment of cancer cells with SSZ diminishes the levels of GSH within cells, leading to excessive ROS.

Secondly, inhibition of xCT is associated with ferroptosis induction. Firstly, described in 2012, ferroptosis is a form of iron-dependent cell death and is characterized by lipid peroxidation and dysregulated mitochondria [7,8]. Lipid peroxide formation is dependent on free labile iron and the amount of ROS present within the cells, which can be influenced by SSZ. Furthermore, GSH is the co-factor of glutathione-peroxidase 4 (GPX4), a main regulator in one of the defense mechanisms against ferroptosis [7,9].

Perturbation of redox homeostasis is an appealing strategy to overcome radioresistance since 67% of the effectiveness of low-linear energy radiation is dependent on the production of ROS. It has already been demonstrated that SSZ can counteract the intrinsic radioresistance of a variety of cancer cells, including glioblastoma and melanoma [6,10]. However, these experiments were performed under normoxic conditions, whereas the radiomodulatory effects of SSZ under hypoxic conditions have not been established before. Hypoxia, a common feature of solid tumors, is still one of the major hurdles in clinical radiotherapy (RT) and, therefore, is of pivotal importance when investigating radiosensitizing approaches [11]. In addition, it has been demonstrated that stress conditions, including hypoxia, increase the upregulation of xCT within cancer cells [6,10].

Recently, a link between RT and ferroptosis has been described. Lei et al., demonstrated that ionizing radiation (IR) increased the amount of ROS within non-small cell lung cancer cells, resulting in enhanced ferroptosis. Additionally, alterations in the expression levels of Acyl-CoA Synthetase Long Chain Family Member 4 (ACSL4) could be observed. ACSL4 is an enzyme required for the synthesis of polyunsaturated fatty acids (PUFAs), which are the main targets of lipid peroxidation, thereby sensitizing the irradiated cells toward ferroptosis. Correspondingly, an increase in the amount of lipid peroxidation was described in patients’ samples after RT compared to matched samples before treatment [9]. Ye et al., investigated the combination of RT with ferroptosis inducers, generating promising results [12]. Both single high-dose RT and fractionated RT triggered ferroptosis, while ferroptosis inhibitors were able to counteract the efficacy of RT [13]. Hence, encouraging evidence reveals that RT induces ferroptosis in a range of cancer cells and patient samples [9,12,13].

Despite the fact that SSZ has drawn a lot of attention over the past decade, no research was conducted on the radiosensitizing effect under hypoxic conditions in CRC. The upregulation of xCT has been linked to poor prognosis in CRC patients, encouraging deeper investigation into the effects of SSZ in this type of cancer [5]. Although one clinical trial is exploring the combination of SSZ and RT in patients suffering from glioblastoma (NCT: 04205357) [14], clinical research regarding this combination remains limited.

In this study, we investigated the radiosensitizing effect of SSZ in human CRC cells and its corresponding mechanisms of action. Subsequent validation of the obtained results was performed in 3D- and xenograft models.

## 2. Materials and Methods

### 2.1. TCGA Colorectal Cancer Cohort Analysis

The mRNA expression levels derived from 594 patients were queried out of the TCGA database, using the cBioPortal website, in the form of transformed z-score data. The analysis compared the *SLC7A11* mRNA expression levels of CRC patients with expression levels in healthy controls. The link between the Ragnum, Buffa, and Winter hypoxia score [15,16,17] and patients with a z-score for *SLC7A11* higher than 2 was directly extracted from the cBioPortal website and downloaded for further analysis.

### 2.2. Cell Lines

The human DLD-1 and HCT116 CRC cells were purchased from the American type culture collection (ATCC, CCL-221, CCL-247). Cells were grown in Roswell Park Memorial Institute 1640 (RPMI) (ThermoFisher Scientific, Merelbeke, Belgium, 21875-034) supplemented with 10% fetal bovine serum (Greiner Bio-One, Vilvoorde, Belgium, FBSEU500).

### 2.3. Treatment

All treatments were performed in RPMI containing HEPES. Sub-confluent cells were treated with SSZ (Sigma-Aldrich, Antwerp, Belgium, S0883) overnight (16 h). ROS scavenger, N-acetyl cysteine (NAC) (10 mM) (Santa Cruz Biotechnology, Heidelberg, Germany, 616-91-1), was added to cultures 1 h prior to and during treatment with SSZ. Ferroptosis inhibitor Ferrostatin-1 (5 µM) (SanBio, Uden, The Netherlands, 17729) was added during treatment with SSZ.

### 2.4. Western Blot

Previously described protocols were used for Western blot analyses [18]. Cell lysates were made by adding a 1% Triton-X lysis buffer to the cells, complemented with a phosphatase inhibitor, protease inhibitor, and leupeptin. Protein concentrations were determined, and equal amounts of protein were loaded on a gel containing 12% polyacrylamide. Nitrocellulose membranes (ThermoFisher Scientific, Merelbeke, Belgium, 88018) were used for an overnight protein transfer at 4 °C. Afterward, blocking of the membranes was conducted with 5% BSA in TBS. Incubation of membranes with primary antibodies was achieved overnight at 4 °C. Near-infrared secondary antibodies (IRDyes 680 RD or 600 CW, Li-Cor Biosciences, Bad Homburg, Germany) were used for labeling primary antibodies. The Odyssey Fc Imaging System (LI-COR, Biosciences, Bad Homburg, Germany) was used for visualization of the signal. The following primary antibodies were used: anti-xCT (Cell Signaling Technology, Leiden, The Netherlands, 12691S) and anti-α-tubulin (Sigma Aldrich, Antwerp, Belgium, T9026).

### 2.5. MTT Assays

Cell viability after SSZ treatment was determined by MTT assay. Cells were treated overnight with a range of doses of SSZ. Afterward, cells were incubated with 50 µL of MTT reagent (0.5 mg/mL) for 1.5 h. Followed by incubation with 200 µL MTT solvent (19:1 DMSO/HCl) in order to homogenize the formazan crystals within the cells. A spectrophotometer (Bio-Rad Laboratories, Temse, Belgium) was used to determine the absorbance at 540 nm. The toxicity profile of SSZ was obtained by division of the absorbance of the treated cells to that of the control cells.

### 2.6. Kinetic Growth Assay

The influence of SSZ on cell growth was determined by following up the confluency of cells in real-time using the Incucyte live cell imager (Essen Biosciences, Ann Arbor, MI, USA). Shortly, cells were grown in 96-well plates and treated with SSZ when they reached sub-confluency. The confluency was measured with the Incucyte software (Incucyte ZOOM 2018A, Essen Biosciences, Royston, UK) for at least 68 h.

### 2.7. Radioresponse Determining Assay

After overnight SSZ treatment, a 6MV Linac was used to irradiate the cells with different doses at a dose rate of 600 cGy/min (Varian Truebeam STx, Palo Alto, CA, USA; BrainLab AG, Feldkirchen, Germany). Clonogenicity was determined by reseeding the cells in 6-well plates as described elsewhere [19]. Cells were incubated for 10–12 days at 37 °C to allow colony formation. Afterward, colonies were visualized by fixing them with crystal violet and were counted. The linear-quadratic model was used to plot survival fractions. Determination of enhancement ratios was performed at the level of 0.1 surviving fractions.

### 2.8. Glutathione Assay

GSH levels were determined using a commercial kit (SanBio, Uden, The Netherlands, 703002) as described elsewhere [20]. Briefly, cells were treated overnight with SSZ. Cells were lysed in ice-cold MES buffer by sonication. Afterward, the supernatant of the different samples (50 µL) was added to the 150 µL assay mixture, and the absorbance was determined at 405 nm using a spectrophotometer over a time range of 30 min.

### 2.9. Thioredoxin Reductase Assay

Thioredoxin reductase (TrxR) activity was determined by using a commercial kit (Sigma-Aldrich, Antwerp, Belgium, MAK409) following the manufacturer’s instructions. Briefly, cells were lysed by sonication in TrxR assay buffer with protease inhibitor. Afterward, 80 µg of each sample was loaded in a 96-well plate, and a 40 µL reaction mix was added. The absorbance was measured following a kinetic method at a wavelength of 412 nm.

### 2.10. Superoxide Dismutase Assay

The activity of superoxide dismutase (SOD) was determined by a commercial kit (Sigma-Aldrich, Antwerp, Belgium, 19160). In short, cells were lysed with a Triton-X lysis buffer supplemented with β-mercaptoethanol (5 mM) and protease inhibitor. Cells were centrifuged at 14,000× *g*, and the supernatant was collected. Next, 20 µL of the supernatant was mixed with a 200 µL assay cocktail and incubated at 37 °C for 20 min. Afterward, the absorbance was measured at a wavelength of 450 nm.

### 2.11. NAD(P)H Levels

The intracellular levels of NAD(P)H were determined by using a commercial kit (AAT Bioquest, Pleasanton, CA, USA, 15291) following the manufacturer’s instructions, as reported elsewhere [21]. Shortly, cells were treated overnight with SSZ. Cells were stained in 250 µL serum-free medium containing 0.5 µL JZL1707 NAD(P)H sensor dye at 37 °C for 1 h. Afterward, cells were analyzed by flow cytometry (FACSCanto, BD Bioscience, Erembodegem, Belgium).

### 2.12. ROS Levels

The quantity of intracellular ROS was determined after SSZ treatment with the ROS assay (Abcam, Cambridge, UK, ab113851), according to the manufacturer’s instructions. Briefly, staining with 5 µM CM-H2DCFDA for 30 min at 37 °C was executed. Stained samples were analyzed by flow cytometry. Incubation of spheroids with 10 µM CM-H2DCFDA for 30 min at 37 °C was carried out, and pictures were taken with the Incucyte ZOOM at a magnitude of 10X (Essen Bioscience, Royston, UK).

### 2.13. DNA Damage Analysis

The extent of DNA damage was determined after irradiation with 8 Gy by staining for γ-H2AX, as described elsewhere [20]. Briefly, cells were treated with SSZ and irradiated. Next, 1 h after irradiation, cells were fixed and permeabilized. Afterward, cells were incubated for 40 min at 4 °C with 0.1 µg γ-H2AX antibody (Abcam, Cambridge, UK, ab228655) and analyzed by flow cytometry.

### 2.14. Ferroptosis Levels

The levels of ferroptosis were determined by using C11BODIPY (ThermoFisher Scientific, Merelbeke, Belgium, D3861). Briefly, cells were incubated with 2 µM C11BODIPY for 15 min at 37 °C and measured by flow cytometry. Spheroids were stained with 4 µM C11BODIPY for 15 min at 37 °C, and pictures were taken with the Incucyte ZOOM at a magnitude of 10X (Essen Bioscience, Royston, UK).

### 2.15. Mitochondrial Membrane Potential

A commercial kit was used to discover the influence of SSZ on the mitochondrial membrane potential (Abcam, Cambridge, UK, ab113852) following the manufacturer’s instructions. Shortly after overnight treatment of the cells with SSZ, staining with 400 nM TMRE dye for 30 min at 37 °C was conducted. Stained samples were analyzed by the use of flow cytometry.

### 2.16. Seahorse Metabolic Profiling

The Seahorse XF96 analyzer (Agilent Technologies, Santa Clara, CA, USA) was consulted for determining the extracellular acidification rate (ECAR) and oxygen consumption rate (OCR). The assay started by seeding 2.5 × 10^4^ cells in 96-well plates. Treatment with SSZ happened overnight. Afterward, cells got equilibrated with unbuffered Dulbecco’s Modified Eagle Medium (DMEM) supplemented with 2 mM glutamine and 10 mM glucose at pH 7.4. Cells were put in a 37 °C and CO_2_-free incubator. After this incubation period, cells were put in the Seahorse analyzer, and specific inhibitors were sequentially added: Oligomycin (1 µM), FCCP (1 µM), and rotenone/Antimycin A (0.5 µM). OCR levels were normalized to the protein content of each condition.

### 2.17. Three-Dimensional (3D) Cell Cultures

Spheroids were generated by seeding 5000 cells of DLD-1 and 4000 cells of HCT116 in 96-well ultra-low attachment plates. Overnight treatment with SSZ was initiated when the spheroids reached a diameter of ±500 µm. The next day, spheroids were irradiated with 8 Gy, and the treatment was removed and replaced with fresh medium. The medium was changed every 3 days, and the growth was followed up by EVOS for 11 days.

### 2.18. In Vivo Tumor Xenograft Model

The Ethical Committee of the Vrije Universiteit Brussel granted approval for the in vivo experiments (VUB—22_552_5). The animal experiment was conducted in accordance with National care regulations. DLD-1 cells (6 × 10^6^) were subcutaneously inoculated into female 6-week-old athymic nude mice (Charles River Laboratories, Miserey, France). After inoculation, DLD-1 xenografts were irradiated as described elsewhere [22]. In short, SSZ was diluted in 0.1 M NaOH and PBS, with an adjusted pH of 7. SSZ (250 mg/kg) was administered intraperitoneally when the tumors reached a size of 200 ± 50 mm^3^ for 5 consecutive days. On day 3 of SSZ treatment, mice were irradiated with a first fraction. Eventually, the mice were irradiated 3 times with 4 Gy using a Linac with an energy of 6 × FFF with a dose rate of 625 cGy/min. A unique CT scan of each mouse was made for situating the lesions and dosimetric determination. The entire duration of the experiment, tumor size and weight was checked with an electrical caliper. Tumor growth was calculated by measuring the tumor volume with the ensuing formula: (length × width^2^) × 0.5.

### 2.19. Data Analysis and Statistics

Analysis of flow cytometric data was executed with FlowJo 7.6.5 Software. Incucyte pictures were analyzed by ImageJ. GraphPad Prism 9.4.0 was used for all the statistical analyses. Except otherwise stated, at least three independent experiments were performed. Data are presented as mean ± SEM. For statistical analyses, One-way ANOVA followed by a Dunnett’s multiple comparison test and Two-way ANOVA followed by a Dunnett’s or Bonferroni multiple comparison test was used. * *p* < 0.05, ** *p* < 0.01, *** *p* < 0.001, **** *p* < 0.0001.

## 3. Results

### 3.1. SLC7A11/xCT Expression Levels Are Linked to Mutational Status of CRC Patients and Correlated with Hypoxic Conditions

Firstly, the expression levels of *SLC7A11* mRNA were investigated (gene encoding xCT) in patients suffering from CRC, extracted out of the TCGA database, and analyzed with the cBioPortal tool for Cancer Genomics. Upregulated levels of *SLC7A11* were present in patients suffering from microsatellite instable (MSI) tumors compared to microsatellite stable (MSS) tumors (Figure 1A). Additionally, the expression level of *SLC7A11* in tumor tissue was compared to the expression in normal tissue. Tumors were divided into two separate groups, mRNA high and mRNA normal, and were linked to three different hypoxia scores. A significant correlation between high *SLC7A11* mRNA levels and two hypoxia scores could be observed; both the Buffa (Figure 1B) and Winter hypoxia score (Figure 1C) correlated with upregulated *SLC7A11*. The third investigated score, namely the Ragnum hypoxia score, did not correlate with *SLC7A11* mRNA levels (Figure 1D). A possible explanation for this discrepancy could be that the Ragnum hypoxia score was developed specifically for prostate cancer and is not validated in other cancer types. Additionally, the Ragnum hypoxia score was limited to 32 genes, while the Buffa and Winter hypoxia scores included 51 and 99 genes, respectively, indicating more comprehensive hypoxia scores.

Subsequently, we evaluated the expression of xCT in human MSI CRC cell lines, namely DLD-1 and HCT116 cells, under normoxic (21% O_2_) and hypoxic (1% O_2_) conditions. Our results demonstrated a clear xCT expression in both cell lines (Figure 1E). Consistent with the data obtained from the cBioportal website, xCT expression was upregulated under hypoxic conditions compared to normoxic conditions (1.3-fold in DLD-1 cells and 1.2-fold in HCT116 cells) (Figure 1F). This could indicate a higher dependence on system xC− within the hypoxic fraction of CRC cells, marking this cystine-glutamate antiporter as an interesting target. Additionally, preliminary data suggest that radiation downregulates the *SLC7A11/* xCT expression levels. Similar results were obtained by Lang et al., in fibrosarcoma cells [13]. Altogether, these data support the use of CRC cell lines as a good model to investigate SSZ in combination with RT to alleviate hypoxic radioresistance.

### 3.2. SSZ Radiosensitized Hypoxic CRC Cells While Exerting No Effect on Normoxic CRC Cells

The toxicity profile of SSZ in DLD-1 and HCT116 cells was determined. SSZ decreased cell viability in a dose-dependent manner in both cell lines, independently of the oxygen levels. IC50 values of 3.2 mM and 3 mM were determined for DLD-1 and HCT116, respectively (Appendix A). Additionally, cell confluency was followed in real-time. Doses of SSZ up to 1 mM did not significantly decrease the growth rate of both cell lines (Appendix A). Since the primary aim of this study was to investigate radiosensitizing properties, doses lower than 1 mM (namely, 100 µM, 300 µM, and 500 µM) were used in subsequent experiments to avoid toxicity originating from SSZ alone.

The effect of SSZ on the radiosensitivity of CRC cell lines was examined under both normoxic and hypoxic conditions. Under normoxic conditions, SSZ exhibited no effect on the radiosensitivity of DLD-1 and HCT116 cells (Figure 2A,C). In contrast, under hypoxic conditions, a clear dose-dependent radiosensitizing effect could be observed in both cell lines, with enhancement ratios (ER) of 1.9 in DLD-1 cells and 1.6 in HCT116 cells (Figure 2B,D) [23]. SSZ overcame hypoxic radioresistance but did not alter the intrinsic radioresistance of both cell lines.

### 3.3. SSZ Inhibits Antioxidants GSH and TrxR, Inducing Disturbed Redox Homeostasis, Leading to Increased DNA Double-Strand Breaks under Hypoxic Conditions

SSZ-induced toxicity has been commonly linked to decreased levels of GSH in cancer cells, leading to aberrant levels of ROS [5]. Consequently, the influence of non-toxic doses of SSZ on GSH levels was examined. SSZ significantly decreased GSH levels in a dose-dependent manner in both cell lines, with complete depletion of GSH in DLD-1 cells following treatment with 500 µM SSZ (Figure 3A). Of note, the baseline GSH levels in HCT116 cells (13.2 µM) were markedly higher compared to DLD-1 cells (6.1 µM), indicating an intrinsic difference in balancing redox homeostasis between both cell lines. To further investigate this, the influence of SSZ on other AOs was determined. It has been established that compensation mechanisms can occur to avoid lethal toxicity when one AO system is compromised [24]. Moreover, a clear difference in ER was present between both cell lines, and differences in AO activity are possibly accountable. A second AO enzyme of interest was thioredoxin reductase (TrxR) since TrxR collaborates in functional redundancy with GSH [24]. A significant dose-dependent decrease in TrxR activity was observed in both cell lines after SSZ treatment, with again an increased depletion in DLD-1 cells compared to HCT116 cells (Figure 3B). In continuation, a third AO enzyme was investigated, namely superoxide dismutase (SOD). No difference in SOD activity was observed after SSZ treatment (Figure 3C). Lastly, we investigated the influence of SSZ treatment on NAD(P)H, the co-factor of GSH and TrxR. A dose-dependent increase in the amount of NAD(P)H was observed in DLD-1 cells, with a significant increase after treatment with 500 µM SSZ (1.5-fold) (Figure 3D). However, this increase was not observed in HCT116 cells (Figure 3D). Theoretically, increased levels of NADPH are present as a compensation mechanism to create more reduced Trx and GSH. Additionally, increased levels of NADH are associated with a dysregulated mitochondrial metabolism [25].

The AO system tightly regulates the levels of ROS present within the cells; hence, altered levels of AO could disrupt redox homeostasis. As a logical next step, the levels of ROS were determined. A dose-dependent increase in the amount of ROS was observed after SSZ treatment in both cell lines (4.1-fold in DLD-1 cells and 6.8-fold in HCT116 cells), reaching statistical significance from 300 µM (Figure 4A). In order to evaluate whether these increased levels of ROS were responsible for the previously observed radiosensitizing effect under hypoxic conditions (Figure 2B,D), cells were treated with SSZ in combination with ROS scavenger NAC and irradiated with 8 Gy. NAC partially reversed the radiosensitizing effect after IR in both DLD-1 and HCT116 cells and induced significant changes after treatment with 300 µM and 500 µM SSZ in DLD-1 cells and 500 µM SSZ in HCT116 cells (Figure 4B).

Lastly, ROS are crucial for the generation of DNA damage following IR [23,26]. Therefore, the amount of DNA double-strand breaks after SSZ treatment was determined. SSZ dose-dependently increased the amount of DNA double-strand breaks (up to 1.5-fold) in both DLD-1 and HCT116 cells, reaching statistical significance with the highest dose of SSZ (500 µM) (Figure 4C). Since NAC only partly reverses the radiosensitizing effect of SSZ, induction of DNA double-strand breaks by increased ROS levels is not the only mechanism responsible for the observed radiomodulatory properties of SSZ.

### 3.4. SSZ Causes Lipid Peroxidation Resulting in Ferroptosis in the DLD-1 Cell Line under Hypoxic Conditions

Since SSZ has been described as a ferroptosis inducer [13], we evaluated the contribution of ferroptosis induction to the radiosensitizing effect. First, the amount of lipid peroxidation, a hallmark of ferroptosis, was determined. It became evident that SSZ increased the levels of lipid peroxidation in a dose-dependent manner, reaching significance starting from 300 µM in DLD-1 cells (up to 2.7-fold), while 500 µM was necessary for HCT116 cells (1.5-fold) (Figure 5A). Recently, IR has been reported to be capable of inducing ferroptosis in cancer cells [9,12,13]. Hence, we investigated the amount of lipid peroxidation after a single high dose of IR (16Gy) in combination with SSZ treatment. IR alone was effective at increasing the amount of lipid peroxidation in both cell lines (4-fold in DLD-1 cells and 3-fold in HCT116 cells). However, the combination with SSZ further elevated the levels of lipid peroxidation in DLD-1 cells (7.6-fold increase at 500 µM), while no additional increase was observed in HCT116 cells (Figure 5B).

Although the measurement of lipid peroxidation is mostly used to investigate ferroptosis, this type of cell death is also characterized by additional features, such as dysregulated mitochondria. To confirm the difference in ferroptosis induction between DLD-1 and HCT116 cells, we aimed to map the mitochondrial activity after SSZ treatment. The mitochondrial membrane potential (ΔΨm) was increased after SSZ treatment in DLD-1 cells (up to 1.7-fold), reaching significance with 500 µM (Figure 5C). In line with the obtained results by measuring lipid peroxidation, no difference in ΔΨm was observed in HCT116 cells (Figure 5C).

As a validation step, cells were treated with a ferroptosis inhibitor, Ferrostatin-1, and the impact on the radiosensitizing effect of SSZ was assessed. Ferrostatin-1 clearly reversed the radiosensitizing effect of SSZ under hypoxic conditions in DLD-1 cells (Figure 5D). No counteracting effect was observed in HCT116 cells, confirming the previously obtained results (Figure 5D).

Considering the observed alterations in NADH levels and ΔΨm after SSZ treatment in DLD-1 cells, we hypothesized that adaptations in mitochondrial metabolism could also increase radiosensitizing properties. Mitochondria are important consumers of oxygen, enhancing the hypoxic state of cancer cells and, therefore, an appealing target for radiosensitizing properties [27]. As a measure of metabolic activity, the oxygen consumption rate (OCR) of SSZ-treated cells was measured by a Seahorse analyzer. No changes in OCR or any other parameters were observed (Appendix A), indicating that the observed changes in mitochondrial membrane potential are likely due to the induction of ferroptosis and do not originate from an altered mitochondrial metabolism.

### 3.5. SSZ Radiosensitizes 3D-Colorectal Cancer Models

To confirm the obtained results, 3D-cancer models (spheroids) were generated. Spheroid growth was followed-up after treatment with SSZ alone (300 µM, 500 µM, and 1 mM) and in combination with 8Gy. Higher concentrations of SSZ were used in order to substantiate drug penetration into the core of the spheroid. No effects could be observed with non-toxic doses of SSZ, while a small growth delay was present after treatment with 8Gy in both DLD-1 and HCT116 spheroids. In combination with SSZ, this growth delay was more pronounced, and dose dependency could be observed (Figure 6A,B).

We validated the mechanisms uncovered in the 2D models and could confirm a clear increase in the amount of ROS after SSZ treatment in DLD-1 and HCT116 spheroids (Figure 6C,D) and an increase in lipid peroxidation in DLD-1 spheroids but not in HCT116 spheroids (Figure 6E,F). These results are fully in line with the 2D results.

### 3.6. SSZ Radiosensitizes DLD-1 Tumor Xenografts

Additionally, the in vitro data were further validated in DLD-1 xenografts. Briefly, nude mice received five consecutive days of i.p. treatment with SSZ (250 mg/kg). On day three of the treatment, the first fraction of radiation was given to the tumor. In total, the DLD-1 xenografts were irradiated 3 times with 4Gy/fraction. Afterward, tumor growth was followed-up until the tumor volume exceeded 1500 mm^2^ (Figure 7A). Radiation alone induced a tumor growth delay of 13 days, while a single treatment of SSZ did not delay tumor growth. The combination therapy induced a significant tumor growth delay of 36 days (Figure 7B). Consequently, the median survival of single-treated mice was extended from 49 days in the control group to 65 days in the irradiated group. In conclusion, adding SSZ to the irradiation led to a synergistic effect, with a prolongation in median survival of 89 days (Figure 7C).

## 4. Discussion

CRC still accounts for 10% of cancer-related mortality worldwide, and the standard of care for locally advanced rectal cancer consists of RT followed by surgical resection. Despite continuous advances in radiotherapeutic treatment options, CRC patients often develop radioresistant tumors resulting in relapse and poor prognosis [28], urging the need to develop novel strategies.

In CRC, xCT upregulation has been associated with disease recurrence, venous invasion, and lymph node metastasis [29]. Therefore, the purpose of this study was to potentiate the efficacy of RT by targeting xCT with SSZ. SSZ, an xCT inhibitor and FDA-approved drug, was used to facilitate the translation toward a clinical setting in CRC patients. Since both intrarectal and systemic administration in these patients is feasible and well tolerated.

Our results demonstrated that SSZ enhances the radioresponse in human CRC cells under hypoxic conditions but not under normoxic conditions, as opposed to results published in glioblastoma and melanoma [30,31]. Tumor hypoxia is recognized as one of the principal causes of clinical radioresistance [23]. As a result, modification by hyperbaric oxygen and nitroimidazoles have been exploited to overcome hypoxic resistance. Nevertheless, every effort remained largely ineffective for clinical application [32]. Alternating pharmacokinetics are reported for SSZ with maximal achievable plasma levels between 16 and 300 µM [33,34], suggesting the validity of the acquired concentrations in this study. Recent studies showed elevated levels of ROS present under hypoxic conditions, leading to increased AO in order to cope with this phenomenon. The generation of these elevated AO levels is, among others, dependent on the presence and activity of xCT [35,36]. Pursuant to these findings, we identified a correlation between upregulated expression levels of *SLC7A11* and higher hypoxia scores. Next to this, xCT was upregulated at the protein level when hypoxia was induced. xCT overexpression has been proven to be implicated in chemoresistance against both 5-fluorouracil and cisplatin in CRC. These chemotherapeutic agents are used as first-line treatment in CRC patients in combination with RT [1,37]. Therefore, SSZ treatment can potentially enhance more than solely the radiosensitivity of CRC cells.

It has previously been published that xCT inhibition decreases the levels of GSH and increases the levels of ROS in multiple cancer types, inducing anticancer effects [38]. In this study, we presented similar findings, with increased levels of ROS and decreased GSH levels in both cell lines. However, SSZ did not only decrease the levels of GSH but also the levels of the second most prominent AO, TrxR. Dual targeting of the redundant GSH and TrxR AO systems is perhaps necessary to fully disturb redox homeostasis and, consequently, enhance radioresponses [39,40].

It is well-known that under hypoxic conditions, hypoxia-inducible factors (HIF-1α and HIF-2α) become active. The induction of ferroptosis under hypoxic conditions sparked a lot of controversy in the literature. HIF-1α is described as a negative regulator of ferroptosis through the inhibition of PUFA synthesis and upregulation of *SLC7A11*. On the contrary, HIF-2α is described as a positive regulator of ferroptosis by stimulating the synthesis of PUFA, increasing the intracellular levels of ROS, and elevating the levels of iron transport within the cells [41]. In accordance with the findings of Singhal et al., our results confirmed that ferroptosis is present in CRC cells under hypoxic conditions. This demonstrates the vulnerability of hypoxic CRC cells to ferroptosis [42]. Further investigation into the expression of genes within the HIF family was beyond the scope of this project. Additionally, induction of ferroptosis after IR has been described in a variety of cell lines and patient samples [9,12,13], indicating the significance of ferroptosis within IR. This further corroborates the potential of combination therapy with SSZ, which could synergize at the level of cell death.

Important to mention is that cystine can be derived intracellularly via the transsulfuration pathway, independently of system xC-. However, this contribution is limited and expected to be insufficient to overcome xCT inhibition [43]. Next to the *SLC7A11*-GSH-GPX4 ferroptosis defense system, two other ferroptosis defense systems have been described, namely the ubiquinol pathway and the tetrahydrobiopterin pathway. Targeting these pathways is worth investigating alone or in combination with SSZ. Unfortunately, the exact contribution to ferroptosis resistance of these mechanisms is still unknown [44].

Our findings suggest that SSZ was more potent to increase responses to RT in the DLD-1 cell line compared to the HCT116 cell line. A clear difference in the reduction of both GSH and TrxR after SSZ treatment could be observed between DLD-1 and HCT116 cells, with a more substantiated decrease in DLD-1. Secondly, the observed differences in radioresponses could be linked to the individual activity of xCT on the different cell lines. xCT seems to be more active on the HCT116 cell line compared to the DLD-1 cell line, while both cell lines are MSI. We performed similar radiation experiments with SSZ in MSS CRC cell lines which resulted in an even greater reduction of the observed radiosensitizing effects. To prove this hypothesis, higher concentrations of SSZ are required to reach the same amount of potency in the HCT116 cell line. However, this could not be further examined in an in vitro setting due to the observed antiproliferative effects at higher SSZ doses.

Moreover, SSZ was potent at inducing ferroptosis in DLD-1 cells but not in HCT116 cells. Literature suggests that the sensitivity of CRC cells to ferroptosis can be associated with their p53 status. TP53 is described as negatively regulating ferroptosis in CRC cells, which is in sharp contrast with the function of p53 in other cancer types, where it promotes ferroptosis [45]. Our initial results are in line with this observation since DLD-1 is characterized by mutated p53, while HCT116 is characterized as wild-type P53 [46]. Although this finding should be further validated, the p53 status could be an interesting biomarker for selecting patients who would benefit from ferroptosis-stimulating treatments.

Lastly, our in vitro results were confirmed in human DLD-1 xenografts. Since this model deprived the natural tumor environment, possible immunogenic effects induced by SSZ were lacking. Further research should be performed since activated T-cells are shown to suppress xCT, hence adding to the observed effects of SSZ [13,47].

In this study, we have solely focused on the inhibition of xCT as the mechanism of action of SSZ in CRC. By extension, we consider the present study’s finding highly relevant for patients with low and mid-rectal cancers. SSZ may be administered as an enema to these patients in an attempt to improve the complete clinical response during (total) neoadjuvant chemoradiation. Besides, it is important to keep in mind that SSZ also has anti-inflammatory effects, considering its daily use as a drug with anti-arthritic activity and its use in IBD. The exact anti-inflammatory working mechanism still needs to be elucidated; however, SSZ is assumed to inhibit prostaglandin synthesis [48]. Importantly, prostaglandin inhibition was already linked to radiosensitization three decades ago [49]. Within the clinic, a prime safety concern with the use of SSZ could be the induction of ROS to lethal levels within not only tumor cells but healthy cells as well. Multiple studies have suggested that healthy cells remain unaffected by xCT inhibition since the expression levels of this antiporter are significantly lower compared to tumor tissue and the presence of healthy and robust AO systems [30,31]. In contrast, preliminary experiments showed upregulation of ROS levels in human dermal fibroblasts upon SSZ treatment, albeit not to lethal levels with the concentration range used in this study. In order to bypass these possible toxicity issues in the clinic, local administration of SSZ in the rectum could be beneficial [50].

## 5. Conclusions

Altogether, we have demonstrated that SSZ radiosensitizes hypoxic CRC cells by decreasing the levels of GSH and TrxR, leading to elevated levels of ROS. These elevated ROS levels induced a higher amount of DNA double-strand breaks. Induction of ferroptosis could be beneficial for radiosensitization purposes; however, its contribution could vary between tumors of different genetic backgrounds. Consequently, since the pharmacokinetics and pharmacodynamics of SSZ are well established, the translation of these results into a clinical setting is optimistic.

## Figures and Tables

**Figure 1 cancers-15-02363-f001:**
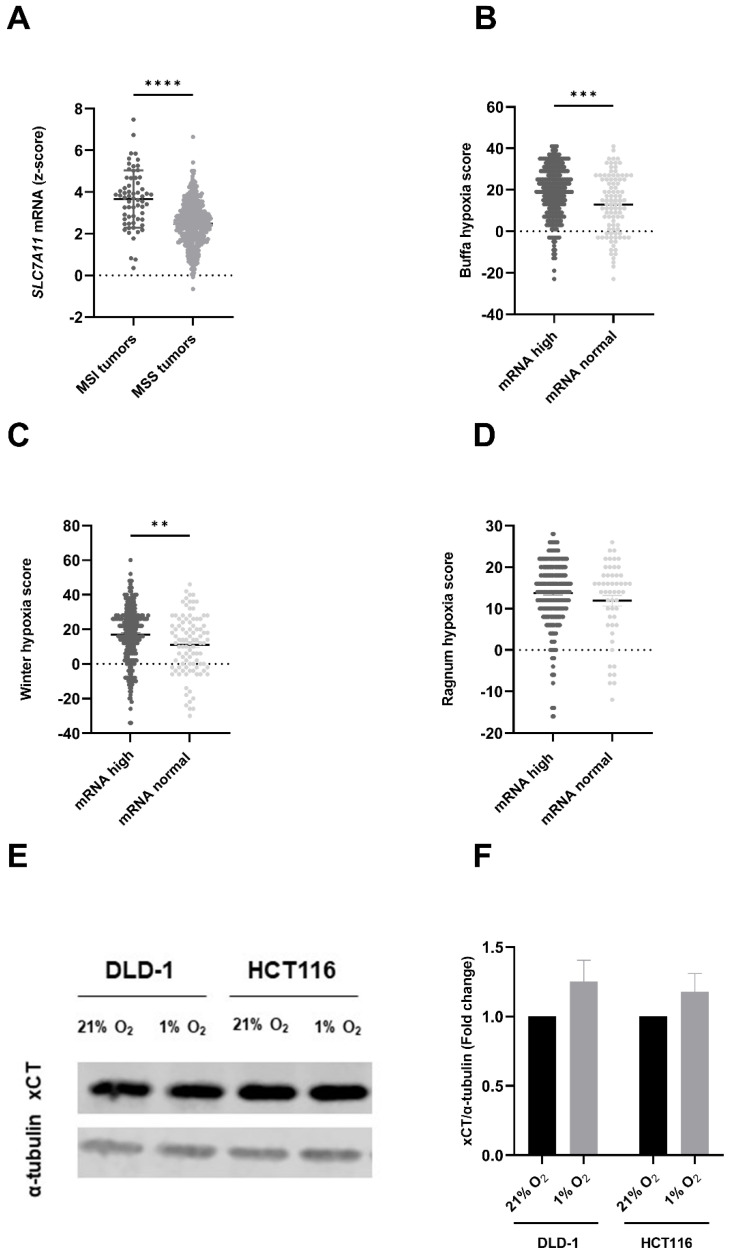
*SLC7A11*/xCT is overexpressed in human colorectal cancer. (**A**) *SLC7A11* mRNA expression level in microsatellite instable (MSI) CRC patients vs. microsatellite stable (MSS) CRC patients. Correlation of *SLC7A11* expression with the (**B**) Buffa hypoxia score, (**C**) Winter hypoxia score, and (**D**) the Ragnum hypoxia score. (**E**) Representative blot showing the expression levels of xCT in DLD-1 and HCT116 cell lines under normoxic (21%) and hypoxic (1%) conditions, using alpha-tubulin as the reference gene. (**F**) Summarizing graph showing the quantitative values of the western blot. Data are shown as mean ± SEM, *n* > 3. An unpaired *t*-test was used in the case of a normal distribution (panel (**A**)), while a Mann Whitney test was used if normality was not present (panels (**B**–**D**)), or a two-way ANOVA followed by Bonferroni’s test (panel (**F**)) was used for statistical analyses. ** *p* < 0.01, *** *p* < 0.001, **** *p* < 0.0001. The uncropped blots are shown in Appendix A.

**Figure 2 cancers-15-02363-f002:**
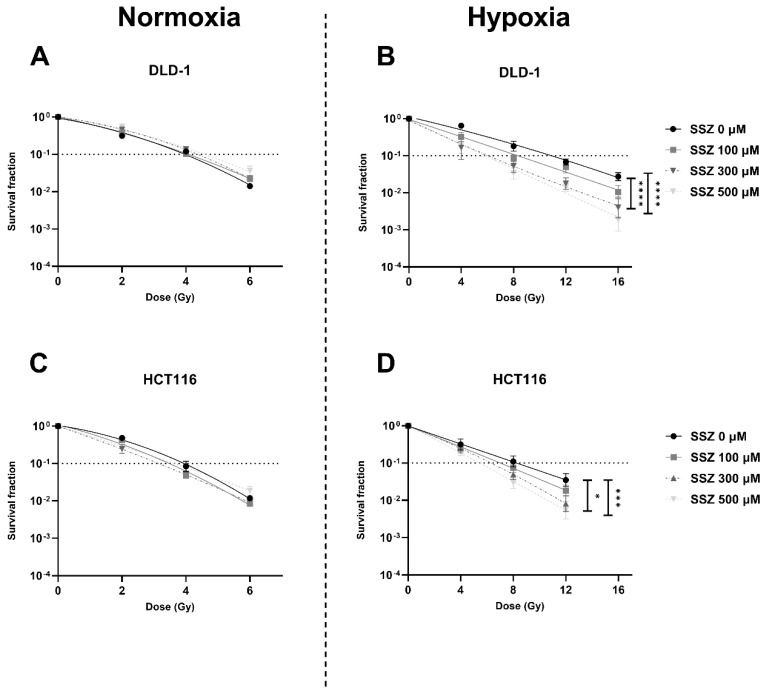
Inhibition of xCT radiosensitizes human hypoxic colorectal cancer cells. DLD-1 and HCT116 were treated with different concentrations of SSZ (16 h) and irradiated with the indicated doses. The radiosensitizing effect was evaluated by colony formation assays. (**A**,**B**) Dose-response curve of the DLD-1 cell line under normoxic conditions (21% O_2_) (**A**) or hypoxic conditions (0.1% O_2_) (**B**). (**C**,**D**) Dose-response curve of the HCT116 cell line under normoxic conditions (21% O_2_) (**C**) or hypoxic conditions (0.1% O_2_) (**D**). Data are shown as mean ± SEM, *n* ≥ 3. A two-way ANOVA followed by Dunnett’s multiple comparison test was used to calculate statistics. * *p* < 0.05, *** *p* < 0.001, **** *p* < 0.0001.

**Figure 3 cancers-15-02363-f003:**
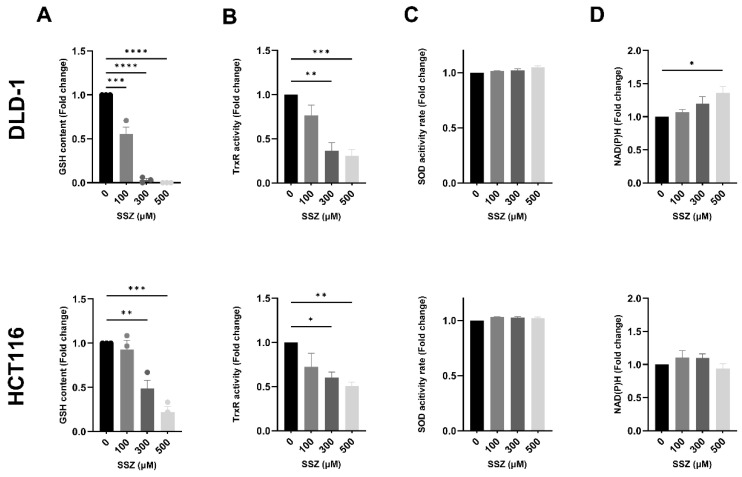
xCT inhibition severely affects different aspects of redox homeostasis under hypoxic conditions. DLD-1 and HCT116 were treated with SSZ (16 h) under hypoxic conditions (1%), and levels of different antioxidants were determined and normalized to their respective controls. (**A**) Summarizing graph showing the levels of GSH in DLD-1 (top) and HCT116 (bottom) cell lines. (**B**) TrxR activity in DLD-1 (top) and HCT116 (bottom) cell lines. (**C**) SOD activity levels were determined in DLD-1 (top) and HCT116 (bottom) cell lines. (**D**) NAD(P)H levels were evaluated in DLD-1 (top) and HCT116 (bottom) cell lines. Data are shown as mean ± SEM, *n* = 3. A one-way ANOVA was performed, followed by Dunnett’s multiple comparison test for statistical analyses. * *p* < 0.05, ** *p* < 0.01, *** *p* < 0.001, **** *p* < 0.0001.

**Figure 4 cancers-15-02363-f004:**
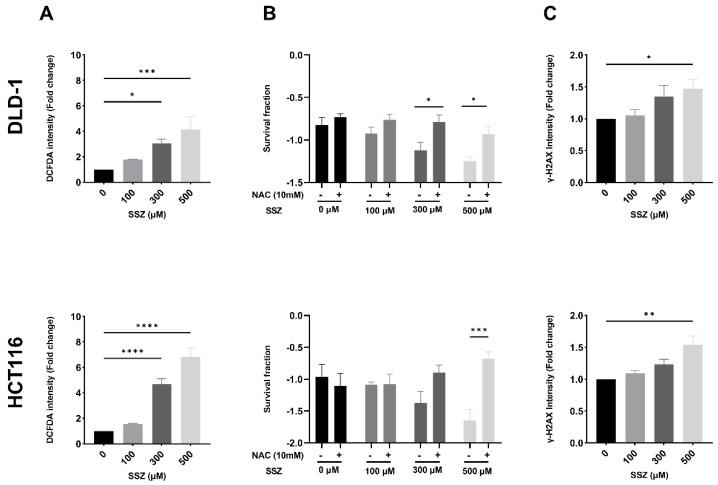
SSZ treatment increases the levels of ROS in hypoxic colorectal cancer cells, leading to DNA damage and radiosensitization. DLD-1 and HCT116 cells were treated (16 h) under hypoxic conditions (1%) with the indicated concentrations. When applicable, NAC was added 1 h prior to SSZ treatment. (**A**) Summarizing graph showing ROS levels in DLD-1 (top) and HCT116 (bottom) cell lines. (**B**) Survival fraction of cells after radiation with 8Gy in DLD-1 (top) and HCT116 (bottom) cell lines. (**C**) The amount of DNA double-strand breaks in DLD-1 (top) and HCT116 (bottom) cell lines. Data are represented as mean ± SEM, *n* ≥ 3. A one-way ANOVA followed by Dunnett’s multiple comparison test or a two-way ANOVA followed by Dunnett’s, or Bonferroni’s multiple comparison test was performed for statistical analyses. * *p* < 0.05, ** *p* < 0.01, *** *p* < 0.001, **** *p* < 0.0001.

**Figure 5 cancers-15-02363-f005:**
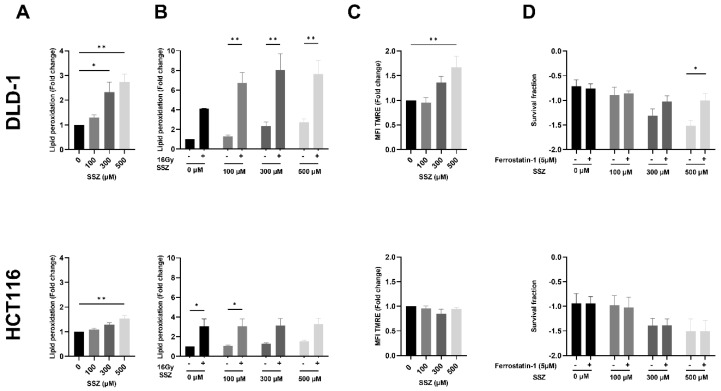
Synergistic effect of SSZ and RT on ferroptosis induction in the DLD-1 cell line under hypoxic conditions. DLD-1 and HCT116 cells were treated with SSZ/Ferrostatin-1 (16 h) under hypoxic conditions (1%), and the induction of ferroptosis was determined. (**A**) The amount of lipid peroxidation after SSZ treatment was evaluated using the C11BODIPY dye in DLD-1 (top) and HCT116 (bottom) cell lines. (**B**) The amount of lipid peroxidation after SSZ treatment + RT (16Gy) 16 h after RT was determined in DLD-1 (top) and HCT116 (bottom) cell lines. (**C**) Mitochondrial membrane potential determination by a TMRE staining and flow cytometry was performed in DLD-1 (top) and HCT116 (bottom) cell lines. (**D**) The survival fraction after RT with 8Gy + SSZ and Ferrostatin-1 treatment was determined in DLD-1 (top) and HCT116 (bottom) cell lines. Data are represented as mean ± SEM, *n* ≥ 3. A two-way ANOVA followed by a Dunnett’s or Bonferroni multiple comparison test, or a one-way ANOVA followed by a Dunnett’s multiple comparison test, was performed for statistical analyses. * *p* < 0.05, ** *p* < 0.01.

**Figure 6 cancers-15-02363-f006:**
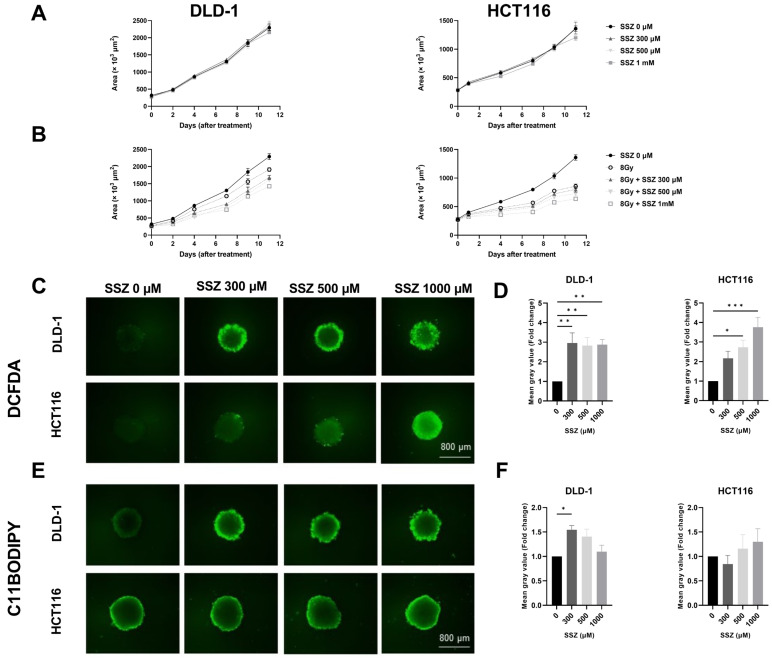
SSZ radiosensitizes colorectal cancer spheroids in a similar manner observed in 2D cultures. DLD-1 and HCT116 3D models were treated with SSZ (16 h) and irradiated with 8 Gy. (**A**) The growth of the spheroids was followed up by the EVOS after SSZ treatment (16 h) in DLD-1 (left) and HCT116 (right) 3D models under hypoxic conditions (0.1%). (**B**) Spheroid growth was followed up after SSZ treatment (16 h) and irradiation with 8 Gy in DLD-1 (left) and in HCT116 (right). (**C**) Representative pictures of the levels of ROS under hypoxic conditions (1%) in DLD-1 (top) and in HCT116 (bottom). (**D**) Summarizing graphs showing the mean gray values in DLD-1 (left) and HCT116 (right). (**E**) Representative pictures of the amount of lipid peroxidation after SSZ treatment under hypoxic conditions (1%) in DLD-1 (top) and in HCT116 (bottom). (**F**) Summarizing graphs showing the mean gray values in DLD-1 (left) and HCT116 (right). Data represented as mean ± SEM, *n* ≥ 3. A one-way ANOVA followed by Dunnett’s multiple comparison test was performed for statistical analysis. * *p* < 0.05, ** *p* < 0.01, *** *p* < 0.001.

**Figure 7 cancers-15-02363-f007:**
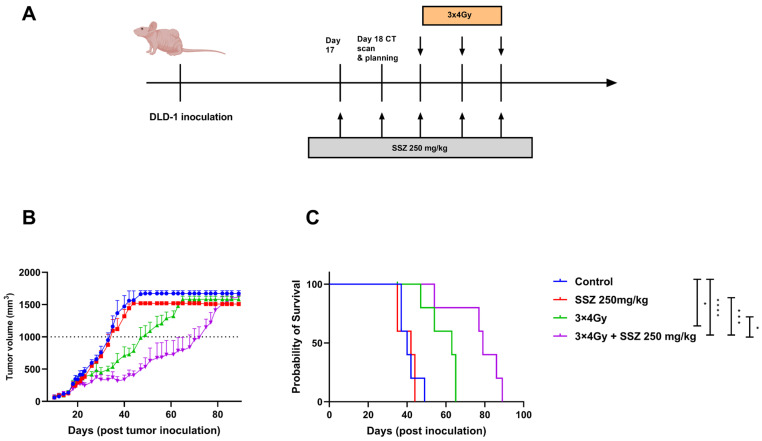
Sulfasalazine synergizes with radiation in DLD-1 xenografts. (**A**) Experimental setup for the treatment of the DLD-1 xenografts in vivo. (**B**) Tumor growth curve, day 0 displays the inoculation of the DLD-1 tumor cells. (**C**) Survival curves of mice, inoculation of the DLD-1 tumor cells is depicted as day 0. Data are shown as mean ± SEM. Results are shown of 1 representative experiment of 2 repeats (*n* = 5). A one-way ANOVA followed by Dunnett’s multiple comparison test was performed for statistical analyses. * *p* < 0.05, *** *p* < 0.001, **** *p* < 0.0001.

## Data Availability

Publicly available data generated by others were used by the authors (TCGA database).

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
