# Peer review of "Repurposing Sulfasalazine as a Radiosensitizer in Hypoxic Human Colorectal Cancer"

_cancers, 2023, doi:10.3390/cancers15082363_

Round 1

Reviewer 1 Report

This exciting study explores the potential benefit of sulfasalazine as a radiosensitizer in hypoxic human colorectal cancer. The paper is well-written and documented, the methods are adequately used, and the conclusions are entirely consistent with the evidence and arguments presented by the results. The research's central question is that in human colorectal cancer cell lines, HCT116 and DLD-1 sulfasalazine enhances radiosensitivity in hypoxic but not normoxic conditions. Thus, sulfasalazine has a potential clinical benefit in overcoming the radioresistance of hypoxic colorectal cancers. Furthermore, the research proposes potential mechanisms for overcoming radioresistance. The topic of the study is original and relevant in the field because no other studies are exploring the potential radiosensitizing effect of sulfasalazine under hypoxic conditions in colorectal cancer. Thus, the paper adds significant scientific value to the current literature approaching the same topic. The paper also includes up-to-date, relevant references to the field, and the results of the present study are pertinently referred to the current findings in the Discussion part. Although the extensive manuscript contains many figures, it is relatively easy to read and understand.

Few concerns:

Radiotherapy is a cornerstone for treating mid-portion and inferior rectal cancers. The benefit of radiotherapy in colonic cancer and upper rectal cancer is limited. Thus, do the authors consider the present study's findings only relevant for rectal or colorectal cancer?

How do the authors consider that the results of the present study could be transposed in clinical practice, and what patients with rectal cancer would benefit from treatment?

Reviewer 2 Report

In this study, Kerkhove et al investigated the potential of sulfasalazine (SSZ), an FDA-approved drug, to augment the efficacy of radiation-induced killing of CRC cells under hypoxic conditions. Using TCGA data, the authors demonstrated that SLC7A11 transcripts (encoding xCT, the target of SSZ) were upregulated in MSI CRCs and elevated levels correlated with hypoxia scores. Hypoxia induced upregulation of xCT was seen in the CRC lines HCT116 and DLD1 in comparison to cells cultured under normoxic conditions. The authors then demonstrated that SSZ treatment sensitized radiation-dependent cell viability in a dose dependent manner under hypoxia. They then conducted a series of experiments to investigate the underlying mechanism for this effect and implicated differences in redox homeostasis, ROS production and induction of ferroptosis. No effects on the mitochondria were noted. Finally, they found that SSZ radiosensitizes CRC lines grown as spheroids and in an in vivo tumor xenograft model.

Overall, this is an important study as the role of SSZ in targeting cancers under hypoxia has not been investigated. The manuscript is well-written, presented data that support conclusions and their interpretations and used the appropriate statistical analyses in their experiments. Moreover, the discussion presented rationale for why SSZ worked more effectively in DLD1 vs HCT116 lines. Only a few points require additional attention to improve the manuscript. 

1. In reference to Figure 1, please comment on why high SLC7A11 mRNA levels did not correlate with the Ragnum hypoxia score, while it does when using the Buffa and Winter scoring systems. What is unique about the Ragnum scoring system that could account for this discrepancy?

2. In Figure 1E, the effects of hypoxia on xCT levels in both DLD-1 and HCT116 cells is modest. The authors should examine SLC7A11 transcript levels in this experiment. In addition, the authors should further investigate whether xCT protein levels, and SLC7A11 transcripts, are increased in both cell lines after radiation under hypoxic conditions.

3. Provide data or a reference supporting the statement that there is a clear difference in ER in both cell lines (line 320).

4. Although the discussion is comprehensive, it would be useful to include consideration of how the doses of SSZ used in their tumor xenograft model correlates with the FDA-approved doses used for the treatment of RA, IBD, or perhaps more appropriately, the referred trial in glioblastoma.   
